# Patient-Reported Outcomes after Laser Ablation for Bladder Tumours Compared to Transurethral Resection—A Prospective Study

**DOI:** 10.3390/cancers16091630

**Published:** 2024-04-24

**Authors:** Nina Nordtorp Deacon, Ninna Kjær Nielsen, Jørgen Bjerggaard Jensen

**Affiliations:** 1Department of Urology, Aarhus University Hospital, 8200 Aarhus, Denmarkbjerggaard@skejby.rm.dk (J.B.J.); 2Department of Clinical Medicine, Aarhus University, 8000 Aarhus, Denmark

**Keywords:** non-muscle-invasive bladder cancer, treatment, patient-reported outcomes, TURBT, transurethral resection, laser ablation, side effects

## Abstract

**Simple Summary:**

Superficial bladder cancer is a common disease. The standard method for treatment is a transurethral resection of bladder tumour (TURBT). This is a surgery under general anaesthesia with a complication rate of up to 26%, and it is potentially associated with severe side effects. A newer method, transurethral laser ablation (TULA), is a less invasive procedure performed under local anaesthesia and with a lower risk of complications. We aimed to compare these different transurethral procedures in the treatment of bladder tumours to evaluate any clinically relevant differences in symptoms and side effects. We used questionnaires regarding urinary symptoms, postoperative side effects, and quality of life. We showed that patients undergoing TURBT reported a more extensive early symptom burden and had a higher need for contacting the healthcare system compared to TULA-treated patients. If some TURBTs can be replaced with TULA, it will be beneficial for both future patients and the healthcare system.

**Abstract:**

The standard procedure for diagnosis and treatment of bladder tumours, transurethral resection of bladder tumour (TURBT), is associated with a complication rate of up to 26% and potentially has severe influence on patient-reported outcomes (PRO). Outpatient transurethral laser ablation (TULA) is an emerging new modality that is less invasive with a lower risk of complications and, thereby, possibly enhanced PRO. We collected PRO following transurethral procedures in treatment of bladder tumours to evaluate any clinically relevant differences in symptoms and side effects. This prospective observational study recruited consecutive patients undergoing different bladder tumour-related transurethral procedures. Patients filled out questionnaires regarding urinary symptoms (ICIQ-LUTS), postoperative side effects, and quality of life (EQ-5D-3L) at days 1 and 14 postoperatively. In total, 108 patients participated. The most frequently reported outcomes were postoperative haematuria and pain. Patients undergoing TURBT reported longer lasting haematuria, a higher perception of pain, and a more negative impact on quality of life compared to patients undergoing TULA. TURBT-treated patients had more cases of acute urinary retention and a higher need for contacting the healthcare system. Side effects following transurethral procedures were common but generally not severe. The early symptom burden following TURBT was more extensive than that following TULA.

## 1. Introduction

Bladder cancer (BC) is the 6th most common cancer in Denmark [1], and the 9th most common worldwide [2]. The majority of patients are male, and approximately 75% present with disease confined to the mucosa or submucosa: non-muscle-invasive bladder cancer (NMIBC) [3].

Transurethral resection of bladder tumour (TURBT) is the standard procedure for diagnosis and treatment of NMIBC [4], with complication rates ranging from 5.1% [5] to over 26% [6] primarily involving minor complications, such as transient haematuria, pain/bladder spasm, infection, minor bladder perforation treated conservatively, and affected urination [5,7,8,9].

The chronic nature of NMIBC, with recurrence rates of 24–61% within one year [10] and a stringent invasive follow-up regimen consisting of long-term surveillance with cystoscopies and potentially several TURBTs [11], makes it highly relevant to establish the patients’ perception of how they are affected by this treatment. Despite TURBT being a frequently performed operation within the urological speciality, research into patient-reported outcomes (PRO) in the weeks following this procedure is limited.

In recent years, transurethral laser ablation (TULA) has gained attention as a cost-effective alternative to minimise perioperative morbidity associated with the treatment of NMIBC [12]. For primary bladder tumours, TURBT remains the recommended treatment to ensure correct staging, but in the treatment of recurrent tumours, using outpatient transurethral laser techniques under local anaesthesia (LA) reduces the number of invasive procedures with general anaesthesia (GA). Complication rates are low, and the procedure shows good efficacy, though there is a need for further research [13,14].

To identify the best-tolerated treatment option for NMIBC and to enable better alignment of expectations with patients in the future, this study aims to assess PRO of both TURBT and TULA. This study will investigate whether there is a clinically relevant difference in early symptoms after TURBT compared to TULA and examine whether these symptoms differ significantly from cystoscopy without any intervention.

## 2. Materials and Methods

### 2.1. Patients

This prospective, observational study enrolled eligible patients undergoing bladder tumour-related transurethral procedures at the Department of Urology, Aarhus University Hospital (AUH), Denmark, in the period from November 2022 to April 2023. Inclusion took place either at the Department of Day Surgery when patients were in the recovery department, where all consecutive eligible patients were offered participation, or at the Urological Outpatient Clinic on selected days following control cystoscopy or laser ablation.

The treatment modality was based on tumour and patient characteristics, and decided by the primary treating physician who discovered the tumour tissue. Patients were offered participation after their procedure was conducted, and the decision of treatment modality was thus not under the influence of the investigators of this study. General criteria for TULA include recurrent bladder tumours with papillary/non-solid appearance and a diameter < 3 cm. Primary bladder tumours are only considered for TULA in cases where the patient is not eligible for general anaesthesia (GA) due to comorbidity.

The following criteria had to be met for inclusion: age ≥ 18 years, ability to fully comprehend the information provided, and having undergone TURBT, TULA, or follow-up cystoscopy without intervention (‘cystoscopy only’).

Exclusion criteria were as follows: use of a permanent indwelling urinary catheter (IUC), inability to read or understand Danish, a known neurological comorbidity that could influence bladder function, e.g., Parkinson’s disease, or cognitive impairment. Patients undergoing reTURBT or cystoscopy with biopsy in GA at the Department of Day Surgery were removed from final analyses.

Patients were able to participate more than once if they had several surgeries in the given period, as each procedure was registered under a new record ID. Patients were excluded from the study if consent was withdrawn or if no questionnaires were answered. Submission of at least one questionnaire was sufficient for participation.

### 2.2. Procedures

TURBT was performed in GA. A resectoscope (Charriere 32 diameter) was inserted via the urethra to the bladder, and all tumour tissue was resected piecemeal with a bipolar wire loop and sent to pathology.

TULA was performed in LA: 60 mL of a 20 g/L lidocaine solution was administered through a urethral catheter 1 h prior to ablation. Tumour tissue in the bladder was evaporated with a thulium-fibre laser through a flexible cystoscope (Charriere 16 diameter) after a biopsy had been conducted. The laser setting was pulse energy 0.5 Joule, 10 Hz, equal to 5 Watt. The TULA procedure was only performed by experienced urologists familiar with the use of the laser.

Cystoscopy only was performed with a flexible cystoscope (Charriere 16 diameter) after urethral insertion of local analgesic lidocaine gel (2%). The narrow-band imaging (NBI) technique was used routinely. The procedure constituted a control group for comparison with the transurethral procedures with instrumental intervention.

### 2.3. Data Collection

PRO were obtained through a set of three different questionnaires at two time points: ICIQ-LUTS to evaluate urinary symptoms, Patient-Reported Outcome of Transurethral Operation (PROTO) to assess side effects, and EQ-5D-3L to evaluate health-related quality of life (HRQoL). Patients received the first set of questionnaires (regarding the first 24 h) on the day of their procedure. They completed the questionnaires at home during the first postoperative day, and on day two they were contacted by telephone to go through the answers. All telephone interviews were carried out by the same study personnel. The second set of questionnaires (assessing symptoms during the first two weeks) was sent and answered electronically or by post to be completed by the patient on day 14.

### 2.4. Questionnaires

ICIQ-M/F-LUTS [15,16] comprises 13 questions for males (categories: frequency, nocturia, voiding, and incontinence) and 12 questions for females (categories: filling, voiding, and incontinence). A higher score indicates a greater impact of the symptom in question for the patient, though conversely for MLUTS question 6.

EQ-5D-3L [17] is a generic 5-dimensional questionnaire assessing non-disease-specific HRQoL in terms of mobility, self-care, usual activities, pain/discomfort, and anxiety/depression. Higher scores indicate a lower HRQoL.

PROTO (Patient-Reported Outcome of Transurethral Operation) is a questionnaire created for the purpose of this study. It was designed to assess postoperative outcomes of transurethral procedures, and addresses postoperative haematuria, catheterisation, contact with the healthcare system, urinary tract infection (UTI), and postoperative pain, including an 11-point numeric rating scale (NRS) for patients’ pain perception ranging from 0 (free from pain) to 10 points (worst pain imaginable). The questionnaire was formed based on a literature search carried out on PubMed on frequently reported symptoms and side effects from transurethral procedures. The questionnaire underwent a face validity test and semi-structured interviews on a small sample size of patients before being used.

### 2.5. Demographic and Intraoperative Data

Patient demographics, type of surgery, and tumour characteristics were retrieved from electronic patient records. Tumour appearance, size, and number of tumours were based on the surgeon’s description and estimate during the given procedure. Tumour size was verified by CT scan, if available. If there was no mention of the tumour appearance or of the extensiveness of tumour tissue, it was registered as ‘Unspecified’.

Tumour stage (UICC 2017 TNM classification) and grade (WHO 2004/2022 classification) [11] were registered as the highest T-stage and grade described in the pathology report. TULA-treated patients that did not have a biopsy taken were classified according to their latest pathology report.

### 2.6. Statistical Analyses

Data were analysed using R version 4.2.2 (R Core Team (2022)) [18]. Descriptive analyses were conducted to describe patient demographics and clinical characteristics. Categorical variables are presented as number and percentage, n (%). Continuous variables are presented as median and interquartile range (Mdn and IQR) for non-normally distributed variables or mean ± standard deviation (M and SD) for normally distributed variables. For comparison of postoperative PRO between the three procedures, either Pearson’s Chi-squared test or Fisher’s exact test was used for categorical variables, and the Kruskal–Wallis rank sum test was used for continuous variables. A Mann–Whitney U test for nonparametric, unpaired variables was conducted to compare NRS and HRQoL scores between two procedures. Statistical significance was set at *p* < 0.05.

### 2.7. Ethics

The study complied with the Danish Code of Conduct for Research Integrity, and data were stored in alignment with The General Data Protection Regulation. All participants received verbal and written information regarding the study. Informed consent was waived due to the study being a questionnaire survey study; thus, completing and returning the questionnaires was considered as granting consent.

## 3. Results

A total of 108 procedures were included in the final analyses, consisting of 62 procedures performed at the Department of Day Surgery (Figure 1a) and 46 procedures at The Urological Outpatient Clinic (Figure 1b).

Following TURBT, seven participants answered only the day 1 questionnaires, and two participants answered only the day 14 questionnaires. Three patients treated with TULA answered only the day 1 questionnaires, and for cystoscopy, five participants answered only one of the questionnaires, with four people answering only the first set.

The overall mean answering time for the second set of questionnaires was 16.8 days.

### 3.1. Baseline Characteristics

The majority of patients were male and had a median age of 74 years (IQR: 67, 78). Tumours were predominantly papillary in appearance, single, small in size, and displayed tumour stage Ta. Patient demographics and procedure characteristics are summarised in Table 1.

### 3.2. Postoperative Outcome

#### 3.2.1. PROTO

The most common side effects from all transurethral procedures during the first 24 h and the first two weeks were postoperative haematuria (52% at day 1 and 54% at day 14) and pain (55% and 47%; Table 2). Overall, 29% of patients reported haematuria beyond the first two days, and for 22% of TURBT-treated patients, it lasted over eight days.

Patients undergoing cystoscopy only were the least likely to report pain (18% at day 1 and 16% at day 14) and TURBT the most likely (72% and 63%). There was a significant difference in NRS scores between patients treated with TURBT and TULA both at day 1 and day 14. NRS scores between TULA-treated and cystoscopy only patients differed significantly on day 1, but not on day 14. The urethra and meatus urethrae combined were where the most patients located their pain regardless of the procedure and time (Figure 2).

Patients treated with TURBT more frequently reported: long duration of haematuria (≥8 days: 22%), acute urinary retention (AUR) within the first 24 h (11.7%), a higher tendency to be discharged with an IUC (60% vs. 4.3–4.5%; Table 2), and a greater need to contact the healthcare system after discharge (Figure 3). Within the first 24 h, the most frequently contacted healthcare providers were the Department of Urology and home care providers. The most common reason for contacting the healthcare system at any time was catheter-related (bag change, clogging, irrigation, or removal). Other reasons included excessive or prolonged bleeding, readmission, and UTI.

#### 3.2.2. EQ-5D-3L and ICIQ-LUTS

Patients undergoing cystoscopy only reported the lowest impact on HRQoL (Table 3). There was a significant difference in HRQoL between TURBT- and TULA-treated patients at day 1 in favour of TULA, but not on day 14 (day 1: W = 957.5, *p* = 0.004837, day 14: W = 567, *p* = 0.8278). There was no significant difference in HRQoL after TULA and cystoscopy only at either time point (day 1: W = 305, *p* = 0.1431; day 14: W = 203, *p* = 0.4518). Figure 4 depicts the dimensions of the EQ-5D-3L and how each procedure scored.

TURBT-treated males scored higher than those undergoing TULA or cystoscopy only in most ICIQ-LUTS categories on both days (Table 3), with nocturia and voiding symptoms imposing the greatest impact. Overall scores did not change much over time. Females scored the highest on filling symptoms regardless of day (overall Mdn = 7 (3.75, 9) and Mdn = 5 (3, 7)), with no significant difference across all procedures (*p* = 0.4 and *p* = 0.3).

## 4. Discussion

Among the 108 patients investigated in this study, the most frequently reported outcomes were postoperative haematuria and pain. Side effects were common, but predominantly mild. The most severe side effect reported in this study was a risk of AUR of approximately 12% within the first 24 h following TURBT. Neither TULA nor cystoscopy only patients were burdened by this.

Previous retrospective studies have established that 27–70% of hospital readmissions following TURBT are due to haematuria [19,20]. In a recent review on outpatient laser treatment, the prevalence of haematuria was aggregated to 1% [21], and of these one study showed minor haematuria for 24 h in 10% of patients (n = 2) [13]. Though the reported timeframe and method of measurement differed, these previous studies reflect what was also established here: TURBT-treated patients are more likely to experience postoperative haematuria than TULA-treated patients, emphasising the potential advantages of TULA in reducing postoperative haematuria complications and hospital readmissions.

In our study, TULA-treated patients reported significantly lower NRS scores compared to TURBT-treated patients, resembling cystoscopy only patients after two weeks with scores below 1 and limited use of painkillers (Table 2). This study was unique in evaluating pain on day 1 and two weeks post-procedure, as most other studies have evaluated pain directly after TULA as a measure of tolerability [12,13,22,23]. Darrad et al. found that 69.7% of patients rated TULA procedure-related pain 0 out of 10 (indicating no difference compared with flexible cystoscopy alone), and they found no significant variables associated with the patients’ pain scores [22]. Considering that the pain reported by patients in our study was primarily localised to the urethra and meatus urethrae (Figure 2), it is conceivable that the endoscope itself imposes a substantial proportion of the pain, making it unavoidable regardless of the procedure. Nonetheless, prioritising procedures associated with minimal pain infliction remains advisable.

More than half (60%) of the TURBT-treated patients in our study were discharged with an IUC, inflicting discomfort and leading to increased healthcare contacts. Only a few patients treated at The Urological Outpatient Clinic had the need for an IUC, and they had little or no need to contact the healthcare system. Although it is a different kind of contact, emergency hospital readmission following TURBT stands at approximately 11% [20], whereas the review by Malde et al. showed that no instances have been reported following TULA [21]. The need for an IUC may be attributed to the resection technique of each procedure, though also tumour size and patient characteristics. The risk of bladder perforation is associated with resection depth and tumour location risks, causing an obturator nerve reflex, and it is a well-recognised complication during TURBT [24]. Contrarily, there has only been one reported case during TULA [21,25]. This is likely attributable to the laser not causing obturator nerve reflexes and to the surgical technique of laser ablation instead of resection. In addition, TULA-treated patients typically have smaller and more superficial tumours (Table 1). These factors result in minor bleeding and fewer bladder perforations, reflected in the reduced need for an IUC at discharge (Table 2). Moreover, patients with a relative bladder outlet obstruction might have an aggravation of this following GA. If TULA can spare patients the discomfort of an IUC and reduce the number of contacts to health services, it would be an advantage for both patients and an already strained healthcare system.

Very few studies have compared TURBT and TULA. To our knowledge, this is the first study comparing short-term PRO between TURBT, TULA, and cystoscopy without intervention as the main endpoint. The primary focus in most studies on TULA has been on the oncological outcome: mortality, recurrence rate, progression-free survival time, and the cost-effectiveness [12,14,22,23]. One study focusing on treatment-related morbidity after outpatient diode laser coagulation and TURBT in the treatment of recurrent bladder tumours was conducted by Pedersen et al. [26]. They found that laser treatment had fewer complications than TURBT (2% vs. 10.1%), and the impact on postoperative QoL assessed seven days after the procedure revealed lower LUTS and worry scores in favour of diode laser coagulation. While the laser ablation technique differs from the method used in our study, the findings of Pedersen et al. are consistent with our results, suggesting that TULA is less burdensome for patients.

An aspect not assessed in this current study is the cost-effectiveness of TULA compared to TURBT. Jønler et al. [27] and Hermann et al. [13] previously conducted comparative cost analyses in a Danish setting. They found that outpatient laser treatment offers cost-saving benefits compared to standard TURBT (a EUR 350 and 1602 reduction, respectively), indicating economic incentives for adopting TULA as a standard treatment for small recurrent bladder tumours. Less invasive procedures, that are also less expensive, would be beneficial, particularly considering that BC is the cancer with the highest lifetime treatment cost per patient [28].

Strengths of this study include the comprehensive comparison across tumour types and procedures, the prospective design of the study, and the high response rate achieved through telephone contact on day 1. A limitation of the study was the use of not properly validated questionnaires, including the PROTO. This questionnaire was created and used due to a lack of a validated questionnaire assessing postoperative complications, such as haematuria, pain, and catheter-related issues. As mentioned, it underwent a face validity test and semi-structured interviews on a small number of patients before being used. Going forward, PROTO should be properly validated to ensure the reliability and comparability of results. ICIQ-LUTS is validated for assessing lower urinary tract symptoms (LUTS) but has not been validated to assess the impact of transurethral procedures on voiding symptoms. However, previous studies have shown that TURBT affects urination by inducing pollakisuria, urge, dysuria, and nocturia [9], so lacking questionnaires validated specifically for this purpose, we considered ICIQ-LUTS a feasible alternative. Additionally, missing baseline information on the patients’ LUTS makes it difficult to attribute symptoms directly to the procedure. To address this limitation, patients included within the last month of our study (n = 29) were asked an additional question on day 14: “Compared to how your urination was before the procedure, is it now: improved/as before the procedure/worse”? Here, 62% reported that LUTS were as before surgery. This was also illustrated by the fact that the overall LUTS scores in our study did not change much over time (Table 3).

Another limitation of the study is the heterogeneity of the study group regarding tumour characteristics, and the uneven distribution of patients undergoing each procedure. This is a consequence of the observational study design.

A concern with TULA is the lack of proper histopathological grading and staging, as the tumour is vaporised and does not undergo pathological examination. In our setting, TULA was primarily used for small recurrent tumours with known pathology from previous TURBTs. For primary tumours, it was only used in frail elderly patients unable to tolerate GA and for whom histological findings would have limited consequences. Nevertheless, it is possible to obtain one or more biopsies before ablation if histopathology is desired. Investigating the histopathological outcome of TULA biopsies compared to TURBT resection tissue, as well as the long-term outcomes of each procedure, will provide a more comprehensive understanding of the advantages and limitations of using TULA compared to TURBT.

## 5. Conclusions

This study provided insight into the outcomes following different bladder tumour-related transurethral procedures. Overall, side effects were common but predominantly mild. The early symptom burden was more extensive following TURBT than TULA. Outcomes from TULA resembled those of cystoscopy only more, compared to TURBT. If the majority of TURBTs for recurrent NMIBC can be replaced by TULA, it will be beneficial for the recovery period of future NMIBC patients.

## Figures and Tables

**Figure 1 cancers-16-01630-f001:**
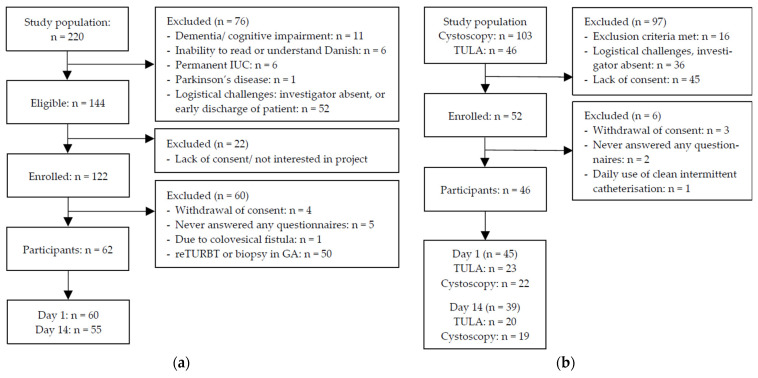
Participant flow at (**a**) the Department of Day Surgery and (**b**) The Urological Outpatient Clinic.

**Figure 2 cancers-16-01630-f002:**
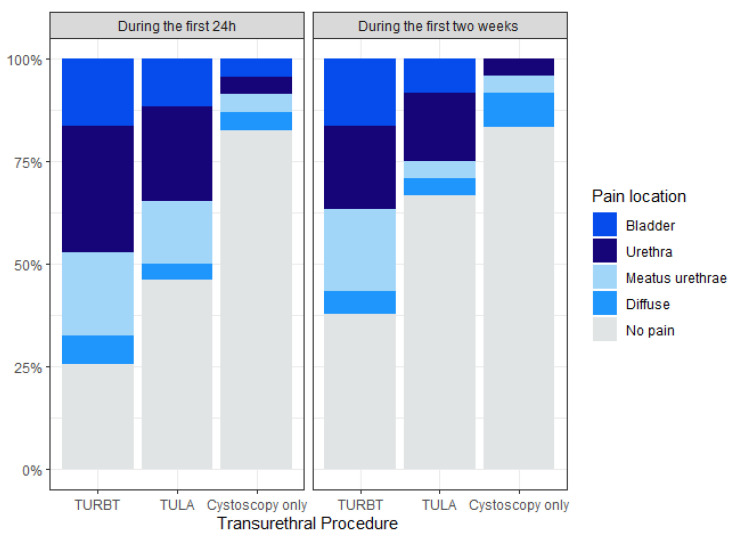
Stacked bar chart displaying pain localisation for each procedure. Note: each patient was able to answer more than one pain location if experiencing pain.

**Figure 3 cancers-16-01630-f003:**
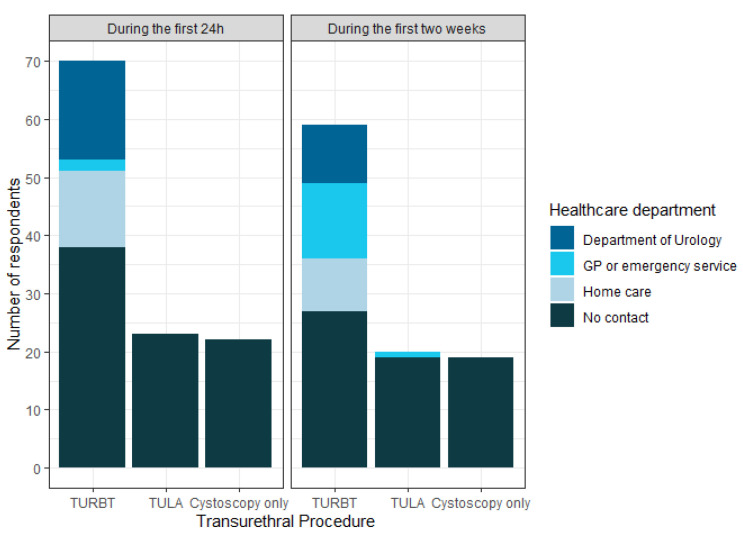
Bar chart displaying contact with the healthcare system for each procedure. Note: each patient was able to answer more than one healthcare department.

**Figure 4 cancers-16-01630-f004:**
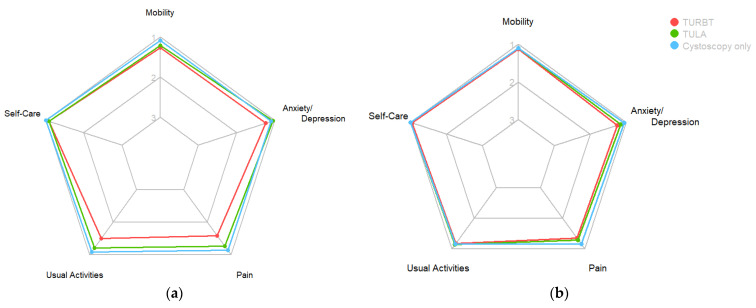
Radar chart of mean option in all EQ-5D-3L HRQoL categories at (**a**) day 1 and (**b**) day 14.

**Table 1 cancers-16-01630-t001:** Patient demographics and procedure characteristics.

	Overall N = 108	TURBT N = 62	TULA N = 23	Cystoscopy Only N = 23
Male sex, n (%)	94 (87%)	51 (82%)	22 (96%)	21 (91%)
Median age (IQR)	74.0 (67.0, 78.0)	70.5 (64.2, 75.8)	77.0 (70.5, 80.5)	76.0 (72.0, 81.5)
Median BMI (IQR)	27.0 (25.0, 29.0)	27.0 (25.0, 29.0)	27.5 (24.2, 28.8)	26.0 (23.0, 29.5)
ASA Classification				
1	22 (20%)	10 (16%)	5 (22%)	7 (30%)
2	55 (51%)	35 (56%)	9 (39%)	11 (48%)
3	28 (26%)	17 (27%)	7 (30%)	4 (17%)
4	3 (2.8%)	0 (0%)	2 (8.7%)	1 (4.3%)
Diabetic	21 (19%)	13 (21%)	4 (17%)	4 (17%)
In anticoagulant treatment	51 (47%)	27 (44%)	12 (52%)	12 (52%)
Previous bladder cancer diagnosis	55 (51%)	14 (23%)	22 (96%)	19 (83%)
Tumour appearance				
No tumour tissue	17 (16%)	1 (1.6%)	0 (0%)	16 (70%)
Papillary	65 (60%)	41 (66%)	19 (83%)	5 (22%)
Sessile	5 (4.6%)	1 (1.6%)	3 (13%)	1 (4.3%)
Solid	16 (15%)	16 (26%)	0 (0%)	0 (0%)
Unspecified	5 (4.6%)	3 (4.8%)	1 (4.3%)	1 (4.3%)
Tumour number				
Single	61 (67%)	44 (72%)	14 (61%)	3 (43%)
Multiple	30 (33%)	17 (28%)	9 (39%)	4 (57%)
Size of largest tumour				
≤1 cm	41 (45%)	22 (36%)	19 (83%)	0 (0%)
2–3 cm	26 (29%)	22 (36%)	2 (8.7%)	2 (29%)
≥4 cm	12 (13%)	12 (20%)	0 (0%)	0 (0%)
Unspecified	12 (13%)	5 (8.2%)	2 (8.7%)	5 (71%)
Tumour stage				
T2-4	9 (8.3%)	9 (15%)	0 (0%)	
T1 (a + b)	11 (10%)	11 (18%)	0 (0%)	
Ta	49 (45%)	31 (50%)	18 (78%)	
CIS	6 (5.6%)	5 (8.1%)	1 (4.3%)	
T0	23 (21%)	0 (0%)	0 (0%)	
Tumour grade				
High	29 (27%)	26 (42%)	3 (13%)	
Low	42 (39%)	28 (45%)	14 (61%)	
Papilloma	1 (0.9%)	1 (1.6%)	0 (0%)	

Data are presented as frequency, n (%), or median (interquartile range (IQR)). BMI: body mass index, ASA: American Society of Anesthesiologists.

**Table 2 cancers-16-01630-t002:** Patient-Reported Outcome of Transurethral Operation (PROTO).

	DAY 1 (during the First 24 h)					DAY 14 (during the First Two Weeks)				
Outcome	Overall N = 105	TURBT N = 60	TULA N = 23	Cystoscopy Only, N = 22	*p* ^1^	Overall N = 94	TURBT N = 55	TULA N = 20	Cystoscopy Only, N = 19	*p* ^1^
Postoperative haematuria	55 (52%)	48 (80%)	4 (17%)	3 (14%)	<0.001	51 (54%)	42 (76%)	7 (35%)	2 (11%)	<0.001
>2 days						27 (29%)	21 (38%)	6 (30%)	0 (0%)	
≥8 days						13 (14%)	12 (22%)	1 (5%)	0 (0%)	
AUR	7 (6.7%)	7 (11.7%)	0 (0%)	0 (0%)		2 (2%)	2 (3.6%)	0 (0%)	0 (0%)	
Catheter when leaving hospital	38 (36%)	36 (60%)	1 (4.3%)	1 (4.5%)	<0.001					
Subsequent catheter						8 (8.5%)	7 (13%)	1 (5%)	0 (0%)	0.3
Pain	58 (55%)	43 (72%)	11 (48%)	4 (18%)	<0.001	44 (47%)	34 (63%)	7 (35%)	3 (16%)	<0.001
Mean NRS score (SD)	3.12 (2.99)	4.42 (2.99)	1.91 (1.86)	0.71 (1.74)	<0.001	1.57 (2.18)	2.20 (2.41)	0.75 (1.02)	0.56 (1.69)	<0.001
Use of painkillers	55 (52%)	48 (80%)	6 (26%)	1 (4.5%)	<0.001	26 (28%)	22 (41%)	2 (10%)	2 (11%)	0.005
UTI						8 (9%)	5 (9%)	1 (5%)	2 (11%)	0.9
Contact with healthcare system	22 (21%)	22 (37%)	0 (0%)	0 (0%)	<0.001	28 (30%)	27 (50%)	1 (5%)	0 (0%)	<0.001

Data are presented as frequency, n (%), or mean (±standard deviation (SD)). NRS scores were treated as normally distributed data. ^1^ *p*-values from Pearson’s Chi-squared test, Fisher’s exact test, or the Kruskal–Wallis rank sum test performed across procedures. AUR: acute urinary retention; NRS: numeric rating scale; UTI: urinary tract infection.

**Table 3 cancers-16-01630-t003:** Results from EQ-5D-3L and ICIQ-LUTS.

	DAY 1 (during the First 24 h)					
Outcome	Overall N Mdn (IQR)	TURBT N Mdn (IQR)	TULA N Mdn (IQR)	Cystoscopy Only N Mdn (IQR)	*p* ^1^	Scoring Range **
EQ-5D-3L HRQoL	105 6 (5, 7)	60 6 (6, 7.2)	23 5 (5, 6.5)	22 5 (5, 5)	<0.001	[5–15]
ICIQ M-LUTS *						
Voiding	55 7 (5, 8.5)	16 8 (6, 9)	20 7 (4.75, 10)	19 5 (4, 6.5)	0.011	[0–20]
Incontinence	56 3 (1, 5)	16 4 (3, 6)	21 2 (1, 5)	19 2 (0.5, 3.5)	0.019	[0–24]
Frequency	57 1 (0, 2)	17 1 (0, 2)	21 1 (0, 2)	19 0 (0, 1)	0.13	[0–4]
Nocturia	56 1.5 (1, 3)	17 2 (1, 3)	20 2 (1, 3)	19 1 (1, 2)	0.7	[0–4]
ICIQ F-LUTS *						
Filling	10 7 (3.75, 9)	7 7 (6.50, 9)	1 1 (1, 1)	2 6.5 (4.75, 8.25)	0.4	[0–16]
Voiding	11 0 (0, 1)	8 0.5 (0, 1.25)	1 0 (0, 0)	2 0.50 (0.25, 0.75)	0.7	[0–12]
Incontinence	10 4 (1, 7)	7 3 (1, 7)	1 5 (5, 5)	2 6 (3.50, 8.50)	0.9	[0–20]
	**DAY 14** **(during the first two weeks)**					
**Outcome**	**Overall** **N** **Mdn (IQR)**	**TURBT** **N** **Mdn (IQR)**	**TULA** **N** **Mdn (IQR)**	**Cystoscopy only** **N** **Mdn (IQR)**	***p* ^1^**	**Scoring range ****
EQ-5D-3L HRQoL	93 5 (5, 6)	55 5 (5, 6)	20 5 (5, 6)	18 5 (5, 6)	0.5	[0–15]
ICIQ M-LUTS *						
Voiding	74 7 (5, 10)	41 8 (6, 11)	17 7 (5, 9)	16 7 (4.8, 8)	0.2	[0–20]
Incontinence	74 3 (2, 5)	41 4 (2, 5)	18 3 (2, 3.75)	15 2 (1.5, 4.5)	0.3	[0–24]
Frequency	78 1 (0, 2)	43 2 (1, 3)	18 1 (0, 2)	17 0 (0, 1)	<0.001	[0–4]
Nocturia	78 2 (1, 3)	43 2 (1.5, 3)	18 1.5 (1, 2)	17 1 (1, 2)	0.042	[0–4]
ICIQ F-LUTS *						
Filling	13 5 (3, 7)	10 5 (3.2, 7)	1 1 (1, 1)	2 7.5 (5.2, 9.8)	0.3	[0–16]
Voiding	13 1 (0, 1)	10 0.5 (0, 1)	1 0 (0, 0)	2 2 (1.5, 2.5)	0.2	[0–12]
Incontinence	13 1 (0, 4)	10 0 (0, 3.25)	1 1 (1, 1)	27.5 (5.25, 9.75)	0.2	[0–20]

Data are presented as the number of respondents (N) and median (interquartile range (IQR)). ^1^ *p*-values from the Kruskal–Wallis rank sum test. * Only patients without an indwelling urinary catheter answered ICIQ-LUTS questionnaires. ** Complete possible scoring range: higher scores indicate worse LUTS or lower HRQoL. HRQoL: health-related quality of life, LUTS: lower urinary tract symptoms.

## Data Availability

The data presented in this study are available upon request from the corresponding author.

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
