# Peer review of "Patient-Reported Outcomes after Laser Ablation for Bladder Tumours Compared to Transurethral Resection—A Prospective Study"

_cancers, 2024, doi:10.3390/cancers16091630_

Round 1

Reviewer 1 Report

Comments and Suggestions for Authors

The authors reported a comparison between cystoscopy, TURB and TULA in terms of symptoms and complication rate, in a prospective series. 

It is hard to understand the rational of the study: how can cystoscopy alone be compared to interventional procedures?

In addition who decided to perform TURB or TULA. The selection criteria are not clear.

Which was the anesthetia protocol? Any difference between TURB and TULA? Furthermore, the male gender with LUTS can impact on results. All these issues should be mentioned as limitations. 

Comments on the Quality of English Language

Acceptable

Author Response

Dear reviewer,
Thank you for taking the time to evaluate our work. We are committed to ensuring the clarity and accuracy of our manuscript and welcome any further suggestions you may have.

Comment 1: It is hard to understand the rational of the study: how can cystoscopy alone be compared to interventional procedures?

Response 1: We thank you for pointing this out. Cystoscopy without intervention works as a control group opposite TULA and TURBT treatment. This has now been specified in the revised manuscript.

Comment 2: In addition, who decided to perform TURB or TULA. The selection criteria are not clear.

Response 2: This is a valid point, and we have now clarified this matter in the revised method section.

Comment 3: Which was the anesthetia protocol? Any difference between TURB and TULA?

Response 3: We thank you for this comment. TURBT was performed in general anaesthesia, and TULA in local anaesthesia (60ml of 20g/L lidocaine-solution) as stated in the method section 2.2 Procedures.

Comment 4: Furthermore, the male gender with LUTS can impact on results. All these issues should be mentioned as limitations.

Response 4: We acknowledge the limitations you have pointed out. Limitations regarding LUTS are mentioned in the end of the discussion section.

Reviewer 2 Report

Comments and Suggestions for Authors

Dear Authors,

I read with interest your patient-reported outcome after laser ablation for bladder tumours compared to transurethral resection – a prospective study. 

Overall, I think the article is of genuine clinical interest.

However, it would be interesting to clarify some points. In particular:

Abstract: It could be useful to briefly explain to readers the inclusion and exclusion criteria in more detail. Simply summary is repetitive and redundant.

Materials and Methods:

·      What was the laser setting used in TULA technique?

·      Which was the experience of surgeons performing the TULA technique?

Discussion:

I suggest discussing the results of:

·      Bladder perforation during transurethral resection of the bladder: a comprehensive algorithm for diagnosis, management and follow-up. C. Lonati at al. DOI: 10.23736/S2724-6051.21.04436-

Author Response

Dear reviewer,
We appreciate the time you have taken to review our manuscript, and kindly thank you for your feedback. Please find the corrections highlighted in the re-submitted manuscript.

Comment 1: Abstract: It could be useful to briefly explain to readers the inclusion and exclusion criteria in more detail. Simply summary is repetitive and redundant.

Response 1: We agree with your assessment. Due to the word-limit we are not able to add more information to the abstract, instead we have specified the inclusion and exclusion criteria as well as the selection criteria in the revised method section.

Comment 2: Materials and Methods:

  • What was the laser setting used in TULA technique?
  • Which was the experience of surgeons performing the TULA technique?

Response 2: Thank you for taking interest in the TULA technique. The laser setting was pulse energy 05. Joule, 10 Hz – equal to 5 Watt. The TULA procedure was only performed by experienced urologists familiar with the use of the laser. This has now been included in the manuscript.

Comment 3: Discussion: I suggest discussing the results of:

  • Bladder perforation during transurethral resection of the bladder: a comprehensive algorithm for diagnosis, management and follow-up. C. Lonati at al. https://pubmed.ncbi.nlm.nih.gov/34263743/

Response 3: We thank you for bringing this article to our attention. We have considered your suggestion and included the article in our revised discussion.

Reviewer 3 Report

Comments and Suggestions for Authors

The authors investigated the level of patient satisfaction following laser ablation (TULA) of bladder tumors compared to transurethral resection of bladder tumors (TURBT). A total of 62 patients who underwent TULA completed the questionnaire on postoperative days 1 and 14. The authors observed that postoperative complications such as pain and hematuria were common but mild in TULA patients. TULA-treated patients reported significantly lower NRS scores compared to TURBT-treated patients. TULA was performed under local anesthesia, while TURBT required general anesthesia.

This pilot study is intriguing as it highlights the safety and tolerability of TULA. The improved subjective outcomes of TULA may be attributed partly to the fact that patients undergoing TULA had superficial tumors, a smaller number of tumors compared to the TURBT cohort, and a reduced frequency of urinary indwelling catheter usage (only one patient left the hospital with a urethral catheter).

The manuscript is well illustrated with 4 figures and 3 tables. It has 24 references.

I have concerns regarding the oncological outcomes and radicality of TULA. However, this aspect falls outside the scope of this manuscript. Additionally, I am concerned about the accuracy of pathological diagnosis following TULA, as biopsy results may not be directly comparable to TUR samples in terms of pathological diagnosis.

Furthermore, the manuscript requires minor proofreading for English language accuracy.

Comments on the Quality of English Language

Although the manuscript is generally well-written, it requires minor proofreading to ensure English language accuracy.

Author Response

Dear reviewer,
Thank you for taking the time to review our manuscript, your feedback is valuable. Please find the corrections in track changes in the re-submitted manuscript.

Comment 1: The improved subjective outcomes of TULA may be attributed partly to the fact that patients undergoing TULA had superficial tumors, a smaller number of tumors compared to the TURBT cohort, and a reduced frequency of urinary indwelling catheter usage (only one patient left the hospital with a urethral catheter).

Response 1: This is a valid observation. TULA-treated patients had smaller and more superficial tumours, but also underwent a more superficial resection with less bleeding and fewer bladder perforations and therefore with reduced needs for catheters. This is discussed in the revised manuscript.

Comment 2: I have concerns regarding the oncological outcomes and radicality of TULA. However, this aspect falls outside the scope of this manuscript. Additionally, I am concerned about the accuracy of pathological diagnosis following TULA, as biopsy results may not be directly comparable to TUR samples in terms of pathological diagnosis.

Response 2: We understand your concern and acknowledge this as a limitation for the usage of TULA. In our setting TULA is almost exclusively used for recurrent tumours, where the patient’s previous pathology is known. It can be used for primary tumours in frail elderly patients who cannot tolerate general anaesthesia, and where the histology would therefore not have much consequence. However, if histology is desired, it is possible to take of biopsy of the tumour tissue before vaporising. The challenges regarding histopathology of TULA are mentioned in the revised manuscript.

Comment 3: Furthermore, the manuscript requires minor proofreading for English language accuracy.

Response 3: Thank you for informing us. The revised manuscript has undergone proofreading before resubmission. Language revision was performed by a bilingual healthcare professional with in-depth knowledge of Urology.

Reviewer 4 Report

Comments and Suggestions for Authors

The article investigates the comparison between TURBT and TULA in terms of patient-reported outcomes, specifically focusing on urinary symptoms, postoperative side effects, and quality of life. This study is significant as it provides valuable insights into alternative treatments for superficial bladder cancer, aiming to reduce complication rates and enhance patient recovery and quality of life.

Areas for Improvement:

  • Validation of Instruments: The study utilized a newly created questionnaire (PROTO) for evaluating postoperative outcomes, which has not been previously validated. Future studies should consider validating this questionnaire or employing validated instruments to ensure the reliability and comparability of results. This soul be considered in the limitation. 
  • One critical consideration not extensively addressed in the study comparing TULA with TURBT is the absence of histological data when opting for TULA over TURBT. Histopathological analysis obtained through TURBT plays a pivotal role in accurately staging and grading bladder tumors, which are crucial for guiding subsequent management, follow-up strategies, and prognostication. The absence of tissue diagnosis with TULA raises concerns about the potential for underestimation of tumor aggressiveness, risk of overlooking muscle-invasive disease, and the inability to detect carcinoma in situ. This limitation significantly impacts the long-term management and follow-up protocol for patients undergoing TULA. This should be discussed in details. 
  • In light of the limitations associated with TULA, specifically its inability to provide histological data for comprehensive cancer staging, recent findings on the urobiome present a promising adjunctive diagnostic approach. This study PMID: 38298766 identifies Porphyromonas somerae in FM urine samples as a specific biomarker for BCa, offering a potential non-invasive method to detect and monitor BCa in patients undergoing less invasive treatments like TULA. Such innovative diagnostic strategies could significantly enhance patient management by enabling risk stratification and personalized follow-up protocols without the need for invasive tissue sampling. This should he included in your discussion along with the suggested ref. 
  • In the same contest, beyond the procedural choices of TULA and TURBT, systemic inflammatory responses, as evidenced by the systemic immune-inflammation index (SII), emerge as significant predictors of oncological outcomes in bladder cancer. According to a retrospective study conducted from 2016 to 2022 (PMID: 38138166), elevated preoperative SII values were associated with worse prognoses in BCa patients undergoing radical cystectomy, including higher rates of lymph node invasion, advanced tumor stages, and poorer survival outcomes. These findings suggest that integrating systemic inflammation markers like SII into preoperative assessment could offer a more holistic approach to patient management, complementing the procedural benefits of less invasive techniques like TULA with crucial insights into the patient's systemic health status and cancer prognosis. Please include and discuss the ref. 
  • Long-term Outcomes: The study focuses on short-term patient-reported outcomes. Investigating long-term outcomes and recurrence rates associated with TULA compared to TURBT could provide a more comprehensive understanding of the benefits and potential drawbacks of each procedure. This should be better highlighted in the limitations and in future prospectives
  • Comparative Cost Analysis: Including a detailed cost analysis comparing TULA and TURBT could provide valuable insights into the economic implications of adopting TULA as a standard treatment, which is particularly relevant given the healthcare system's interest in cost-effective treatments.
  • Addressing Limitations: The article mentions the use of non-validated questionnaires and the lack of baseline data on patients' urinary symptoms. Future research should aim to address these limitations by incorporating validated instruments and baseline symptom assessments to strengthen the study's findings.

Author Response

Dear reviewer,
Thank you for your insightful comments. We appreciate the time you have taken to evaluate our work. Please find the corrections highlighted in the re-submitted manuscript.

Comment 1: Validation of Instruments: The study utilized a newly created questionnaire (PROTO) for evaluating postoperative outcomes, which has not been previously validated. Future studies should consider validating this questionnaire or employing validated instruments to ensure the reliability and comparability of results. This soul be considered in the limitation.

Response 1: We agree. It would be highly relevant to fully validate the PROTO questionnaire in a future study.

Comment 2: One critical consideration not extensively addressed in the study comparing TULA with TURBT is the absence of histological data when opting for TULA over TURBT. Histopathological analysis obtained through TURBT plays a pivotal role in accurately staging and grading bladder tumors, which are crucial for guiding subsequent management, follow-up strategies, and prognostication. The absence of tissue diagnosis with TULA raises concerns about the potential for underestimation of tumor aggressiveness, risk of overlooking muscle-invasive disease, and the inability to detect carcinoma in situ. This limitation significantly impacts the long-term management and follow-up protocol for patients undergoing TULA. This should be discussed in details.

Response 2: We understand your concern and acknowledge this as a limitation for usage of TULA. It has been added to the revised manuscript. In our setting TULA is almost exclusively used for recurrent tumours, where the patient’s previous pathology is known. It can be used for primary tumours in frail elderly patients who cannot tolerate general anaesthesia, and where the histology would therefore not have much consequence. However, it is worth mentioning, that if histology is desired, it is possible to take of biopsy of the tumour tissue before vaporising.

Comment 3: In light of the limitations associated with TULA, specifically its inability to provide histological data for comprehensive cancer staging, recent findings on the urobiome present a promising adjunctive diagnostic approach. This study PMID: 38298766 identifies Porphyromonas somerae in FM urine samples as a specific biomarker for BCa, offering a potential non-invasive method to detect and monitor BCa in patients undergoing less invasive treatments like TULA. Such innovative diagnostic strategies could significantly enhance patient management by enabling risk stratification and personalized follow-up protocols without the need for invasive tissue sampling. This should he included in your discussion along with the suggested ref.

In the same contest, beyond the procedural choices of TULA and TURBT, systemic inflammatory responses, as evidenced by the systemic immune-inflammation index (SII), emerge as significant predictors of oncological outcomes in bladder cancer. According to a retrospective study conducted from 2016 to 2022 (PMID: 38138166), elevated preoperative SII values were associated with worse prognoses in BCa patients undergoing radical cystectomy, including higher rates of lymph node invasion, advanced tumor stages, and poorer survival outcomes. These findings suggest that integrating systemic inflammation markers like SII into preoperative assessment could offer a more holistic approach to patient management, complementing the procedural benefits of less invasive techniques like TULA with crucial insights into the patient's systemic health status and cancer prognosis. Please include and discuss the ref.

Response 3: Thank you for bringing these interesting articles (PMID: 38298766, PMID: 38138166) to our attention. It is indeed exciting how many new less invasive modalities are emerging in bladder cancer treatment, and also beneficial to the patients. However, the study on urobiome in BC patients undergoing TURBT, and the study on systemic immune-inflammation index (SII) as a predictor for the oncological outcomes in patients treated with radical cystectomy falls outside the scope of our study, as it focuses on patient-reported outcomes of transurethral procedures.

Comment 4: Long-term Outcomes: The study focuses on short-term patient-reported outcomes. Investigating long-term outcomes and recurrence rates associated with TULA compared to TURBT could provide a more comprehensive understanding of the benefits and potential drawbacks of each procedure. This should be better highlighted in the limitations and in future prospectives.

Response 4: We acknowledge the limitations you have pointed out and have addressed them in the revised version of the manuscript.

Comment 5: Comparative Cost Analysis: Including a detailed cost analysis comparing TULA and TURBT could provide valuable insights into the economic implications of adopting TULA as a standard treatment, which is particularly relevant given the healthcare system's interest in cost-effective treatments.

Response 5: We thank you for outlining this relevant perspective. Other studies by e.g. Hermann (PMID: 29607745) and Goméz (PMID: 28462594) have analysed the cost-effectiveness of TULA versus TURBT. We have considered your suggestion and added a section on this in the revised discussion.

Comment 6: Addressing Limitations: The article mentions the use of non-validated questionnaires and the lack of baseline data on patients' urinary symptoms. Future research should aim to address these limitations by incorporating validated instruments and baseline symptom assessments to strengthen the study's findings.

Response 6: Thank you for this very valuable comment, we agree and have already stated this in the discussion.

Reviewer 5 Report

Comments and Suggestions for Authors

Dear Editor

The authors examined day one and day 14 patient-reported outcome after TURBT, TULA and cystoscopy. They found that TULA is similar to cystoscopy, and that both are much more tolerable compared to TURBT. They concluded that it would be beneficial to replace some TURBTs with TULA.

Comments:

1.    This is a well conducted and controlled prospective study.

2.    It should be emphasized that TULA does not provide histology and thus is suitable only for small recurrent tumors.

3.    These cases can also benefit from surveillance only.

4.    Complications may be reported according to the CD system.

5.    It is not clear why the authors used a 32 diameter resectoscope. A 24 will do the same job with less damage.

Comments on the Quality of English Language

None

Author Response

Dear reviewer,
Thank you for taking the time to evaluate our manuscript, your comments are valuable to our work. Please find the revisions highlighted in the re-submitted manuscript.

  1. This is a well conducted and controlled prospective study.

Response 1: Thank you very much!

  1. It should be emphasized that TULA does not provide histology and thus is suitable only for small recurrent tumors.

Response 2: This is a valid point, and we acknowledge this as a limitation for the usage of TULA. It has been added to the revised manuscript.

  1. These cases can also benefit from surveillance only.

Response 3: We thank you for this comment, it is a fine point. In our setting, TULA is used for small recurrent tumours, slightly larger than those who are candidates for active surveillance. The selection criteria for TULA have been specified in the revised manuscript. We have chosen to focus on usual standard treatments such as control cystoscopy, TULA and TURBT.

  1. Complications may be reported according to the CD system.

Response 4: This is a valid observation. It would be optimal if complications were collected and classified according to Clavien-Dindo, however the focus of our study was patient-reported outcomes, thus more subjective outcomes not suitable for the CD classification.

  1. It is not clear why the authors used a 32 diameter resectoscope. A 24 will do the same job with less damage.

Response 5: The 32 diameter resectoscope is standard equipment at our department and was therefore used.

Round 2

Reviewer 4 Report

Comments and Suggestions for Authors

worthy of publication